# INTERPRETABLE LATENT DISTRIBUTIONS USING SPACE-FILLING CURVES

## ABSTRACT

Deep generative models are well-known neural network-based architectures that learn a latent distribution whose samples can be mapped to sensible real-world data such as images, video, and speech. Such latent distributions are, however, often difficult to interpret. In generative adversarial networks (GANs), some earlier supervised methods aim to create an interpretable (structured) latent distribution or discover interpretable directions for image editing which require exploiting the data labels or annotated synthesized samples during training, respectively. In contrast, we propose using an unsupervised structured distribution modeling technique that incorporates space-filling curves into vector quantization, which makes the latent distribution interpretable by capturing its underlying morphological structure. We apply this technique to model the latent distribution of pretrained StyleGAN2 and BigGAN networks on various image datasets. Our experiments show that the proposed approach yields an interpretable model of the latent distribution such that it determines which part of the latent distribution corresponds to specific generative factors such as age, pose, hairstyle, background, data class, etc. Furthermore, we can use the points and direction of a space-filling line for controllable data augmentation and applying intelligible image transformations, respectively. The implementation of our proposed method is publicly available[1].

## 1 INTRODUCTION

Generative adversarial networks (GANs) (Goodfellow et al., 2014) are powerful deep generative models that can be applied to a wide range of applications such as data augmentation (Antoniou et al., 2017; Shorten & Khoshgoftaar, 2019), image manipulation (editing) (Härkönen et al., 2020; Voynov & Babenko, 2020; Shen et al., 2020; Jahanian et al., 2019; Yüksel et al., 2021; Shen & Zhou, 2021; Abdal et al., 2021; Wang et al., 2018b; Plumerault et al., 2020; Yang et al., 2021; Goetschalckx et al., 2019; Alaluf et al., 2022; Roich et al., 2022; Pehlivan et al., 2023; Liu et al., 2023), video generation (Wang et al., 2018a), and speech enhancement (Pascual et al., 2017). For image data, GANs map a latent distribution to an output image space by learning a non-linear mapping through adversarial training (Voynov & Babenko, 2020). After learning such mapping function, GANs are capable of creating realistic high-resolution images by sampling from the latent distribution (Karras et al., 2019; 2020; Brock et al., 2018). However, this latent distribution is a black-box in the sense that we cannot interpret what image each latent vector would generate regarding semantic attributes (e.g. gender, age, accessories, etc.) (Shen et al., 2020) and what are the human-interpretable directions for editing these semantic attributes (Voynov & Babenko, 2020). Hence, having a more comprehensive interpretation of the latent space is an important research problem that if solved leads to more controllable generations.

Prior works can be categorized into three principal approaches aimed at making the latent distributions of generative models more interpretable.

**1. Introducing structure into the latent distribution using data labels**. The main rationale behind these approaches (Klys et al., 2018; Xue et al., 2019; An et al., 2021) is that they take advantage of labeled data (with respect to the features of interest) and train the generative model in a supervised manner to learn a structured latent distribution in which data with specific labels reside in isolated

---

[1]The link to our GitHub repository (hidden for double-blind rule).

subspaces of the latent distribution. Hence, this structured latent distribution can be interpretable such that the user could have control over data generation and manipulation with respect to the labels. However, these supervised methods suffer from two main drawbacks. First, they require human labeling, whose cost can increase prohibitively when increasing dataset size (Voynov & Babenko, 2020). Second, they might prevent the latent distribution from learning some intrinsic structures that a human labeler is unaware of (Voynov & Babenko, 2020).

**2. Disentangling the latent distribution dimensions**. These methods (Chen et al., 2016; Higgins et al., 2017; Ramesh et al., 2018; Lee et al., 2020; Liu et al., 2020) strive to train the generative model in an unsupervised way to obtain a disentangled latent distribution. In a disentangled latent distribution, changes in each latent dimension make variations only in one specific generative factor, while keeping the other generative factors unchanged. In other words, these approaches aim to model various generative factors existing in the data to different latent dimensions and make these dimensions (generative factors) independent of each other. Therefore, these methods are interpretable in the sense that they allow control over data generation with respect to the generative factors. However, the downside of the techniques with disentangled distribution is that they are less efficient in generation quality and diversity (Voynov & Babenko, 2020).

**3. Exploring meaningful directions in the latent distribution**. The main goal of these techniques (Härkönen et al., 2020; Voynov & Babenko, 2020; Shen et al., 2020; Jahanian et al., 2019; Yüksel et al., 2021; Shen & Zhou, 2021; Abdal et al., 2021; Wang et al., 2018b; Plumerault et al., 2020; Yang et al., 2021; Goetschalckx et al., 2019; Alaluf et al., 2022; Roich et al., 2022; Pehlivan et al., 2023; Liu et al., 2023) is to find the directions in the latent distribution which lead to intelligible data transformations such as changing the age, pose, hairstyle in face synthesis task. Here the interpretability of the latent distribution refers to the user's control over generation process by manipulating latent vectors along these discovered meaningful directions. Based on their methodology, these approaches can be divided into two categories; supervised (Jahanian et al., 2019; Plumerault et al., 2020; Yang et al., 2021; Goetschalckx et al., 2019; Shen et al., 2020) and unsupervised (Härkönen et al., 2020; Shen & Zhou, 2021; Voynov & Babenko, 2020; Yüksel et al., 2021).

One downside of these supervised methods is that they require a large amount of data sampling (collection) together with the use of pretrained classifiers or human labelers to label the collected data with respect to the user-predefined semantic directions, which are expected to be discovered from the latent distribution (Shen & Zhou, 2021). In addition, these methods are limited to finding only the desired directions which the user is interested in (Voynov & Babenko, 2020). On the other hand, in unsupervised methods of Voynov & Babenko (2020); Yüksel et al. (2021), the user has to choose the hyper-parameter $K$ (the number of semantic directions to discover) for training, where a large value for $K$ results in discovering repetitive directions. Similarly, SeFa (Shen & Zhou, 2021) also requires the user to choose $K$. As GANSpace (Härkönen et al., 2020) applies principal component analysis (PCA) on the latent distribution, the number of examined directions ($K$) is really large (equal to the latent distribution dimension). However, not all PCA directions necessarily correspond to changing an independent and meaningful generative factor. Furthermore, in all these unsupervised methods there is no prior knowledge about what type of specific transformation each of these $K$ discovered directions corresponds to. Hence, the user has to do an exhaustive search over all available $K$ directions to determine which directions are practical and what they refer to.

Space-filling vector quantizer (SFVQ) (Vali & Bäckström, 2023) is a recently introduced approach that helps to make the latent distribution more interpretable. SFVQ is a novel structured distribution modeling technique that incorporates space-filling curves into vector quantization (VQ), where VQ codebook elements play the role of corner points in a space-filling curve. According to the intrinsic structure in the space-filling curves, it is expected that SFVQ captures the existing structure in the latent distribution such that adjacent codebook vectors refer to similar content.

In this paper, we applied the SFVQ method to discover the underlying structure in the latent distribution of the pretrained StyleGAN2 (Karras et al., 2020) and BigGAN (Brock et al., 2018) models for an image generation task using FFHQ (Karras et al., 2019), AFHQ (Choi et al., 2020), LSUN Cars (Yu et al., 2015), CIFAR10 (Krizhevsky et al., 2009), and ImageNet (Deng et al., 2009) datasets. In contrast to the methods belonging to the first and second categories above, our unsupervised proposed method neither needs any human labeling nor puts any constraint on the learned latent distribution. Thus, we avoid any degradation in the generation quality and diversity. Moreover, our method does not need any hyper-parameter tuning like selecting the number of discovered direc-

tions ($K$) as in (Shen & Zhou, 2021; Voynov & Babenko, 2020; Yüksel et al., 2021) or the training loss weight coefficient as in (Voynov & Babenko, 2020). Furthermore, in contrast to the methods of (Härkönen et al., 2020; Shen & Zhou, 2021; Voynov & Babenko, 2020; Yüksel et al., 2021), by using SFVQ we have a clear perception of the discovered direction of change in advance, and we are capable of discovering new and unique directions when the bitrate of SFVQ is high. To explore the structure of the latent distribution and find its interpretable directions, the user is required to study the learned SFVQ only once by observation.

Our experiments demonstrate that our proposed technique makes the latent distribution of Style-GAN2 interpretable in the sense that what type of generations to expect from each part of the latent distribution regarding age, hairstyle, pose, background, accessories (for FFHQ dataset), color, breed, pose (for AFHQ dataset) and class of data (for CIFAR10 dataset). In addition, we observe that the space-filling lines (the lines connecting the space-filling curve's corner points or VQ codebook vectors) are mainly located inside the latent distribution. This property renders numerous meaningful latent vectors favorable for controllable data augmentation by interpolating between adjacent code-book vectors. Furthermore, we observe that the direction of the space-filling line connecting two subsequent codebook vectors can be used as a meaningful direction leading to human interpretable image transformation. The main novelties of this paper are:

- Applying SFVQ to interpret the underlying structure and find interpretable directions of changes in the latent distribution for image generation task using GANs (for the first time).

- Employing SFVQ for controllable data augmentation for images using GANs.

## 2 METHODS

### 2.1 VECTOR QUANTIZATION

Vector quantization (VQ) is a method for data compression (analogous to k-means algorithm (Mac-Queen et al., 1967)) that models the probability density function of a distribution by a set of code-book vectors spreading over the whole distribution space. If the bitrate is $B$ bits, VQ divides the data samples into $K = 2^B$ clusters such that each cluster is represented by a codebook vector as its cluster center. Figure 1(a) shows the VQ applied on a 2D pentagon distribution. Applying VQ means to map all data points within a Voronoi region (cluster) to its codebook vector (cluster center). For an input vector $\boldsymbol{x} \in \mathbb{R}^{1 \times D}$ and codebook matrix $\boldsymbol{C} \in \mathbb{R}^{K \times D}$, VQ is defined as

$$\hat{\boldsymbol{x}} = \underset{C_k}{\arg\min} \|\boldsymbol{x} - C_k\|^2, \quad 0 \leq k < K, \tag{1}$$

so that $\|\cdot\|^2$ refers to the Euclidean distance. $C_k \in \mathbb{R}^{1 \times D}$ is the nearest codebook vector from $\boldsymbol{C}$ to the input vector $\boldsymbol{x}$ regarding the Euclidean distance.

### 2.2 SPACE-FILLING VECTOR QUANTIZER (SFVQ)

Space-filling curves are piece-wise continuous curves created by recursion, and if the recursion repeats infinitely the curve entirely fills a multi-dimensional distribution. The Hilbert curve (Sagan, 2012) is a well-known type of space-filling curve, in which the corner points are determined with a specific mathematical formulation at each recursion step. Motivated by space-filling curves, we can interpret the vector quantization (VQ) function as mapping of input data points onto a space-filling curve, whose corner points are the codebook vectors of VQ. This approach is called the space-filling vector quantizer (SFVQ), through which we can map the input data not only on the codebook vectors but also on the line connecting these codebook vectors. Figure 1 illustrates a 6 bit VQ and SFVQ applied on a pentagon distribution.

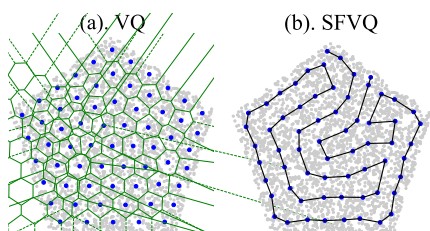

Figure 1: Codebook vectors (blue points) of a 6 bit (a) vector quantizer, and (b) space-filling vector quantizer (curve in black) on a pentagon distribution (gray points). Voronoi regions for VQ are shown in green.

### 2.2.1 SFVQ TRAINING

To train the SFVQ, we use a dithering technique, which guarantees the codebook vectors would not diverge during training. To map the input vector $\boldsymbol{x}$ on the SFVQ's curve, first we generate a dithered codebook matrix $\boldsymbol{C}^{dither}$ by means of interpolation at random places between all two neighboring vectors of the current base codebook matrix $\boldsymbol{C}$. Then, we apply ordinary VQ operation of Equation 1 using the dithered codebook $\boldsymbol{C}^{dither}$ on the input vector $\boldsymbol{x}$. In other words, we map $\boldsymbol{x}$ to the closest codebook vector of $\boldsymbol{C}^{dither}$ as

$$\hat{\boldsymbol{x}} = C_j^{dither} = (1 - \lambda)C_j + \lambda C_{j+1}, \tag{2}$$

such that $C_j$ and $C_{j+1}$ represent the two subsequent codebook vectors from the base codebook $\boldsymbol{C}$ which their interpolation $C_j^{dither}$ is the the closest dithered codebook vector to $\boldsymbol{x}$, and $\lambda$ is the dithering (interpolation) factor. To generate the dithered codebook during training, we sample $\lambda$'s from the uniform distribution of $U[0, 1)$ that ensures random interpolation between subsequent vectors of the base codebook. This type of randomized interpolation imposes a sense of continuity between consecutive vectors of the base codebook and has a regularization effect. If our objective is to train the SFVQ module alone (independent of any other training modules), the mean squared error (MSE) between the input vector $\boldsymbol{x}$ and its quantized form $\hat{\boldsymbol{x}}$ can be applied as the training loss function,

$$\text{MSE}(\boldsymbol{x}, \hat{\boldsymbol{x}}) = \|\boldsymbol{x} - \hat{\boldsymbol{x}}\|^2 = \|\boldsymbol{x} - (1 - \lambda)C_j - \lambda C_{j+1}\|^2. \tag{3}$$

In this scenario, there is no necessity to propagate gradients through the non-differentiable $\arg\min$ function in Equation 1. This is because $\hat{\boldsymbol{x}}$ and consequently the MSE loss are functions of the leaf variables ($C_j$ and $C_{j+1}$) which need gradients. On the other hand, if our intention is to train the SFVQ module together with other modules, we have to find a solution for *gradient collapse* issue (Vali & Bäckström, 2022) to pass the gradients through the non-differentiable $\arg\min$ function. To achieve this, we can apply the recently introduced noise substitution in vector quantization (NSVQ) (Vali & Bäckström, 2022) technique, which defines the final quantized input vector as:

$$\tilde{\boldsymbol{x}} = \boldsymbol{x} + \|\boldsymbol{x} - \hat{\boldsymbol{x}}\| \cdot \frac{\mathbf{v}}{\|\mathbf{v}\|}, \tag{4}$$

where $\mathbf{v}$ is a random vector sampled from a normal distribution ($\mathcal{N}(0, 1)$). NSVQ does not need any hyper-parameter tuning and yields faster convergence and more accurate gradients than the commonly used straight through estimator (Bengio et al., 2013) method.

### 2.2.2 SFVQ INFERENCE

By minimizing the mean squared error (MSE) in Equation 3, the optimal value for $\lambda$ in the interval of $i$ and $i + 1$ can be calculated as

$$\lambda_{optimal} = \frac{(C_{i+1} - C_i)^T(\boldsymbol{x} - C_i)}{\|C_{i+1} - C_i\|^2}. \tag{5}$$

During inference, to determine where the input vector $\boldsymbol{x}$ is mapped on the learned SFVQ line, first, we find the closest codebook vector index $k^*$ from $\boldsymbol{C}$ to $\boldsymbol{x}$ using Equation 1. Here we can inspect two intervals of $\{C_{k^*-1}, C_{k^*}\}$ and $\{C_{k^*}, C_{k^*+1}\}$. Then, we calculate the $\lambda_{optimal}$ and $\hat{\boldsymbol{x}}$ for these two intervals by the use of Equations 5 and 2, respectively. Finally, we compute the MSE between the input vector $\boldsymbol{x}$ and the two obtained mappings $\hat{\boldsymbol{x}}$ for each interval and map $\boldsymbol{x}$ to the interval with the lower MSE value.

## 3 EXPERIMENTS

To evaluate how the space-filling vector quantizer (SFVQ) can be used to explain the underlying structure and interpretable directions of the latent distribution of GANs, we applied it on the intermediate latent distribution ($\mathcal{W}$) of StyleGAN2 (Karras et al., 2020) and first linear layer of BigGAN512-deep (Brock et al., 2018), similar to Härkönen et al. (2020). It has been shown that these layers are more favorable for interpretation because they render more disentangled representations, they are not constrained to any specific distribution, and they suitably model the underlying structure of the real data (Karras et al., 2019; Härkönen et al., 2020; Shen et al., 2020). For Style-GAN2, we employ the pretrained models on FFHQ, AFHQ, LSUN Cars, CIFAR10 datasets, and for

BigGAN we use the pretrained model on ImageNet. We use these pretrained models and extract the latent representations of our desired layers, then we train the SFVQ on the extracted representations.

To inspect the latent distributions from different hierarchy levels, we trained the SFVQ with various bitrates ranging from 2 to 12 bit (4 to 4096 codebook vectors). Similarly to the space-filling curves, we train the SFVQ recursively. Whatever the target bitrate for SFVQ is, we first start with 2 bit SFVQ (only 4 codebook vectors) and then we expand the bitrate from 2 bit to the target bitrate by adding only one bit at each recursion step. Since SFVQ is not very sensitive to hyper-parameter tuning, we adopt a general setup that works for all pretrained models and datasets. In this setup, we trained SFVQ with the batch size of 64 over 100 k number of training batches (for each recursion bitrate) using Adam optimizer with the initial learning rate of $10^{-3}$. We used a learning rate scheduler such that during each recursion bitrate we halved the learning rate after 60 k and 80 k training batches.

## 4 RESULTS AND DISCUSSION

### 4.1 STYLEGAN2: UNIVERSAL INTERPRETATION

To explore a universal interpretation of a latent distribution we apply the SFVQ on that distribution and plot the generated images from obtained SFVQ's codebook vectors (corner points). According to the intrinsic arrangement of corner points in a space-filling curve, we expect the SFVQ to capture a universal morphology of the latent distribution. As the first experiment, we apply the SFVQ on the intermediate latent representation ($\mathcal{W}$) of StyleGAN2 (Karras et al., 2020) pretrained on CIFAR10 dataset. When generating $\mathcal{W}$'s latent vectors during training, StyleGAN2 asks for class label of the image. We sampled the class labels randomly but unbiased against class labels (i.e. we generated same number of latent vectors for each class). In Figure 3, we plot the generated images corresponding to 6 bit SFVQ codebook vectors, i.e. each image corresponds to a codebook vector. At first glance, we observe a clear arrangement with respect to the image class, such that images from an identical category are organized into groups. Also apart from the *horse* class, all animal types and industrial vehicles are located next to each other. In addition, there are some visible similarities for subsequent codebook vectors within a class such as similar object's rotation, scale, color, and background. We also see these observations consistently over different bitrates of SFVQ. Furthermore, when increasing the SFVQ's bitrate by one (doubling the number of codebook vectors), the number of specified codebook vectors for each class will be approximately doubled, and as a result, the proportion of each class remains unchanged. Therefore, from SFVQ (for any bitrate) we can infer the portion each class occupies the latent space. For instance, the *horse* class is always the dominant class of data in the StyleGAN2 latent space by occupying about 25% of codebook vectors.

To inspect the learned SFVQ from another viewpoint, we plotted the heatmap of Euclidean distances between all SFVQ's codebook vectors in Figure 4. We again observe a clear separation between different classes, as each dark box shows a data class. It is important to note that the SFVQ captures this class separation property because of its inherent structure and in a completely unsupervised way. In addition, we spot a bigger dark box shared between *cat* and *dog* classes, because they are the most similar classes and apparently reside close to each other in the latent space. Furthermore, next to the biggest dark box

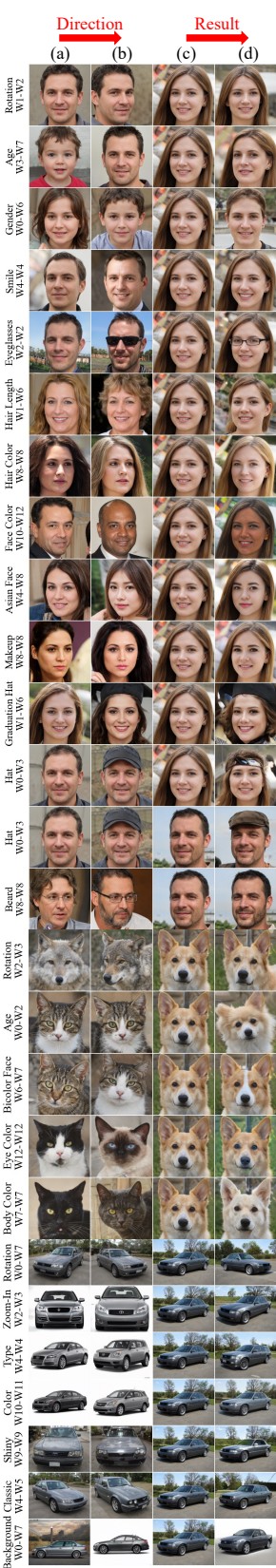

Figure 2: StyleGAN2 interpretable directions.

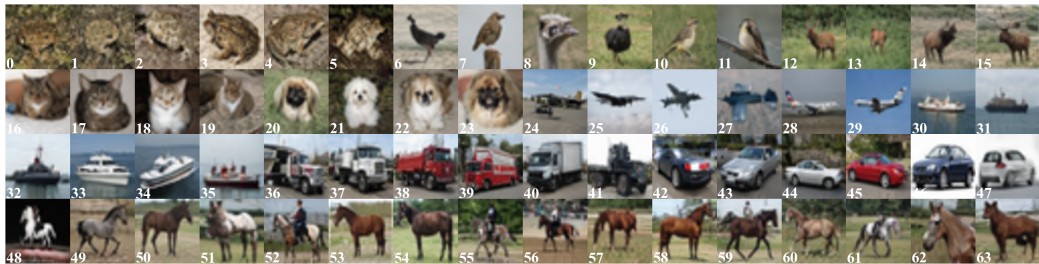

Figure 3: Codebook vectors of a 6 bit SFVQ in $\mathcal{W}$ space of StyleGAN2 pretrained on CIFAR10.

corresponding to *horse*, we see a lighter box which implies that the *horse* class is closer to animal classes (*deer*, *dog*, *cat*, and *bird*) than industrial vehicles, as heuristically expected in reality. We relate the isolation of the *horse* class from other animal classes to the location of initial codebook vectors and the initialization bitrate.

In the second experiment, we applied a 5 bit SFVQ on the $\mathcal{W}$ space of the pretrained StyleGAN2 model on FFHQ dataset. Images corresponding to the SFVQ's codebook vectors are represented in Figure 8. We observe some similarities among neighboring codebook vectors such as baby-aged faces for indices [6-7], hat accessory for indices [13-16], eyeglasses for indices [18-19], rotation from right to left from index 17 to 20, and rotation from left to right from index 27 to 31. Based on our investigations, the StyleGAN2's intermediate latent space for FFHQ, AFHQ, and LSUN Cars are much denser and more entangled than CIFAR10, because all of them are trained on not a very diverse data like CIFAR10. Therefore, that is why we cannot obtain an absolutely clear and distinctive universal interpretation out of the SFVQ's codebook vectors visualization in Figure 8. We provided a similar figure for a 6 bit SFVQ for AFHQ dataset in section A.1 in the Appendix.

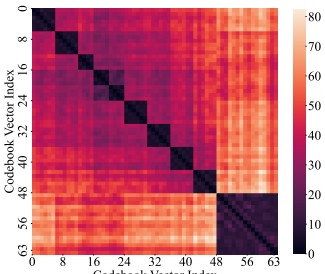

Figure 4: Heatmap of codebook vectors distances for a 6 bit SFVQ in $\mathcal{W}$ space of StyleGAN2 for CIFAR10.

As the third experiment, we examined a 2 bit SFVQ applied on the $\mathcal{W}$ space of StyleGAN2 pretrained on FFHQ dataset and displayed the generated images corresponding to each four SFVQ's corner points in Figure 5(a). We observe a clear separation for females and males while we only have two individual identities each representing the average face for females and males. From this SFVQ curve we can infer some more interesting properties. We hypothesize that each SFVQ line corresponds to a semantic direction of change which is shown in Figure 5(b). Direction I (direction from codebook vector 1 to 2) is for changing rotation to the right, direction II refers to the gender change, and direction III is for changing rotation to the left. For more clarification, we compute the angles between these directions in degrees, which confirms our hypothesis. The direction II is almost orthogonal to two other directions and also directions I and III are approximately inverse of each other (with 159.6 degrees difference).

## 4.2    STYLEGAN2: INTERPRETABLE DIRECTIONS OF CHANGES

Figure 5(a) also reminds us of the PCA-based method of GANSpace (Härkönen et al., 2020) which finds PCA directions as interpretable directions of changes. Similar to the first two PCA directions of GANSpace that refer to the change of gender and rotation, the SFVQ lines are also laid down along the directions in which the training data has the most variance, i.e. gender and rotation. However, in the learned $\mathcal{W}$ distribution, the semantic directions are not necessarily orthogonal to each other, and that is why only the first 100 (out of 512) GANSpace's PCA orthogonal directions lead to noticeable changes. In contrast in the SFVQ curve, each direction could potentially work for a meaningful and obvious change. In addition, by plotting and observing the corner points of SFVQ in advance, we have prior information about the direction of change. Hence, we can simply select the intended direction and only search for the proper layers in StyleGAN2 or BigGAN to modify along the desired direction. Whereas in unsupervised methods of Voynov & Babenko (2020); Yüksel et al. (2021); Shen & Zhou (2021); Härkönen et al. (2020), the user is required to

exhaustively search over all $K$ detected directions to find the desired one (apart from searching for the suitable layers to modify). Therefore, in this case, the SFVQ is more helpful by reducing the search space. All above-mentioned observations and discussions motivate us to utilize the SFVQ curve for discovering interpretable directions of changes which we study in the following.

We applied SFVQ with different bitrates on the $\mathcal{W}$ space of StyleGAN2 pretrained on FFHQ, AFHQ, and LSUN Cars datasets and spotted some useful interpretable directions which are shown in Figure 2. Columns (a) and (b) represent the discovered direction from two SFVQ's subsequent codebook vectors, column (c) is the test vector in the latent space to which we apply the direction, and column (d) is the final result after applying the direction.

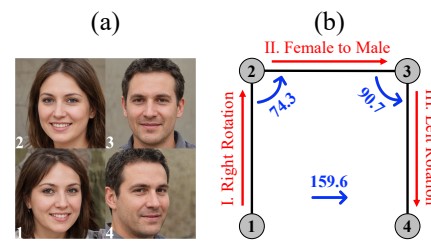

Figure 5: (a) Codebook vectors of a 2 bit SFVQ in $\mathcal{W}$ space of StyleGAN2 pretrained on FFHQ and (b) their semantic directions. Numbers in blue show the angle between directions.

Similar to the GANSpace naming convention, the term W$i$-W$j$ means we only manipulate the style blocks within the range [$i$-$j$]. We take the directions only from SFVQ's subsequent codebook vectors, but not from two necessarily similar though far apart codebook vectors because otherwise, one can find directions accidentally by taking two codebook vectors from an ordinary VQ which might lead to a meaningful direction. To show the practicality of the discovered directions better, we apply all of them only on one identical test image (except for the *Beard* direction which is specified to males). One great advantage of our proposed method over other previous approaches is that it almost keeps the identity of the test image (column (c)) fixed when applying the desired direction. Another major advantage is that we could find many new and unique directions which were not found in previous methods, such as *Asian Face*, *Graduation Hat*, *Hat*, *Beard* for FFHQ, *Age*, *Eye Color*, *Bicolor Face* for AFHQ, and *Classic*, *Background Removal* for LSUN Cars. Important to note that the directions for the AFHQ dataset are class-agnostic, i.e. the direction for one animal works for other animal species, as we found the directions from *Wolf* and *Cat* classes, but we applied them to a *Dog* class. In a similar way, you can see how the *Hat* direction (discovered for males) works logically but in a different way for females. However, some discovered directions do not necessarily work for all animal species in the AFHQ dataset, because the transformations are restricted by the dataset bias of individual animal classes (Jahanian et al., 2019) (see section A.2 in the Appendix).

## 4.3 BigGAN: INTERPRETABLE DIRECTIONS OF CHANGES

BigGAN512-deep (Brock et al., 2018) samples a random vector $\mathbf{z}$ from a normal prior distribution $p(\mathbf{z})$ and maps it to an image. Since in BigGAN512-deep the intermediate layers also take the latent vector $\mathbf{z}$ as input (i.e. skip-$\mathbf{z}$ connections), the latent vector $\mathbf{z}$ has the most effect on the generated output image. Hence, we have to find the semantic directions in $p(\mathbf{z})$ space. However, as $p(\mathbf{z})$ is an isotropic distribution, it is difficult to find useful directions from it (Härkönen et al., 2020). Therefore, similar to GANSpace, we first train the SFVQ on the first linear layer ($\mathcal{L}$) of BigGAN512-deep to search for interpretable directions within this space, and afterward we transfer these directions back to $\mathbf{z}$ space. To this end, we sample $10^6$ random vectors from $p(\mathbf{z})$ and map these vectors to the SFVQ codebook vectors (already trained on $\mathcal{L}$). Finally, for a codebook vector in $\mathcal{L}$, we find its corresponding codebook vector in $p(\mathbf{z})$ by taking the mean of the vectors in $p(\mathbf{z})$ which get mapped to this SFVQ codebook vector. In this way, we would get the corresponding SFVQ curve in $p(\mathbf{z})$ space. Now, we use this SFVQ (in $p(\mathbf{z})$ space) to interpret the latent space of BigGAN512-deep. Note that to compute the SFVQ curve for BigGAN, we select a class label and keep the class vector fixed.

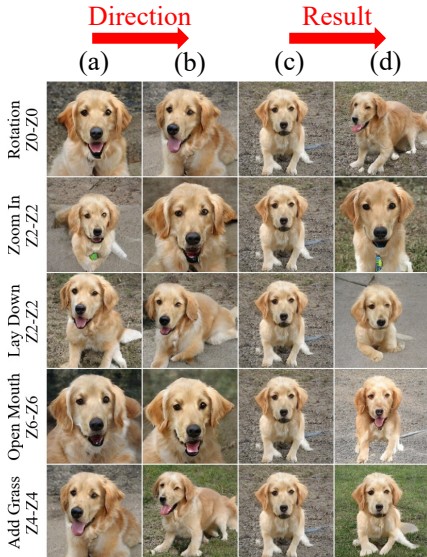

Figure 6: BigGAN interpretable directions.

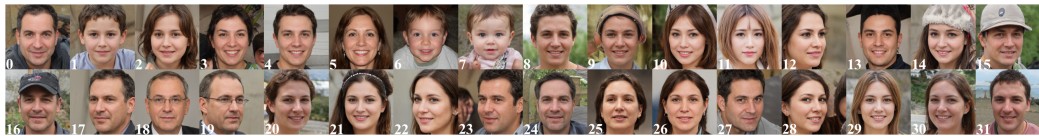

Figure 8: Codebook vectors of a 5 bit SFVQ in $\mathcal{W}$ space of StyleGAN2 pretrained on FFHQ.

We computed the SFVQ curve over different bitrates in the $p(\mathbf{z})$ space of BigGAN512-deep for *golden retriever* class and discovered some interpretable directions which are shown in Figure 6. The directions are not restricted only to these represented here, as one user can find other directions by searching over all SFVQ corner points over different bitrates. Columns (a) and (b) represent the discovered direction from two SFVQ's subsequent codebook vectors, column (c) is the test vector in the latent space to which we apply the direction, and column (d) is the final result after applying the direction. Similar to the GANSpace naming convention, the term Z$i$-Z$j$ means we only manipulate the skip-$\mathbf{z}$ connections within the range $[i\text{-}j]$. Apart from basic geometrical directions (*Rotation* and *Zoom In*), we could also discover some more specific directions such as *Lay Down* and *Open Mouth* as found in Yüksel et al. (2021), and *Add Grass* as found in Härkönen et al. (2020). It is noteworthy that the discovered directions by SFVQ for *golden retriever* class are class-agnostic, i.e. they work also for other classes. For more details see section A.3 in the Appendix.

### 4.4 Comparison with GANSpace

In Figure 7, we compare our proposed method with GANSpace (Härkönen et al., 2020) over five interpretable directions. To have a fair comparison, for GANSpace's edits, we used their official implementation. In addition, we got standard deviation ($\sigma$) values for all directions from GANSpace's method, except for the *Smile* direction which had a small $\sigma$ that led to minimal image edits. The $\sigma$ value determines the magnitude of the step that we take toward a direction of change. We decided to change the $\sigma$ value in a wider range (up to $4\sigma$) to measure the extent of validity for each direction. The images in red squares refer to the initial test image to which we apply the changes. According to the *Gender* direction, we observe that our method remains in the valid range of generations better than GANSpace in changing a female to a male. Regard-

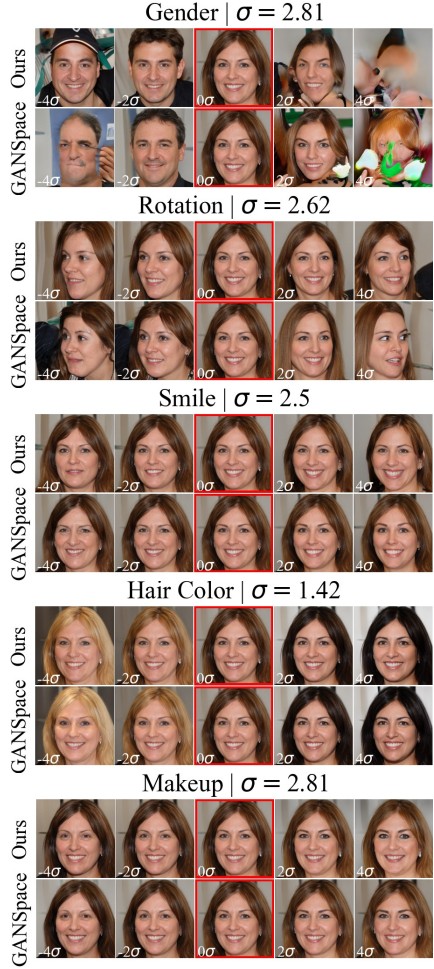

Figure 7: Semantic directions comparison of our method with GANSpace for StyleGAN2 pretrained on FFHQ.

ing the *Rotation* direction, our method is better in keeping the identity, hair color, hairstyle, and lips' shape intact. Under *Smile* direction, our method yields much better smile edits and less changes in face appearance than GANSpace. For *Hair Color* and *Makeup* directions, both methods perform almost the same, except that GANSpace changes the face shininess when changing the hair color.

## 5 Subsidiary Study: the Traveling Salesman Problem

Space-filling vector quantizer (SFVQ) has some parallels with the classic *traveling salesman* problem (TSP) (Flood, 1956) in bringing order and arrangement to a set of codebook vectors. One could ask whether we can achieve a better codebook arrangement than SFVQ by applying an ordinary vector quantization (VQ) as usual and, afterward, use one of the *traveling salesman* solutions to

reorganize VQ codebook vectors. The scenario of TSP is that we have a list of cities (codebook vectors) and the distances between them, then we aim to discover the shortest possible route to visit each city only once. In fact, TSP is an NP-hard problem to solve. We can interpret these cities as the codebook of a VQ. If we learn an 8 bit VQ as usual and we want to rearrange the codebook vectors for minimum distance, then there are 256! possible permutations for rearrangement. This is an astronomically large number ($8.5 \times 10^{506}$). It is thus practically infeasible to do an exhaustive search for all possible permutations in most, relatively high bitrate, cases of VQ. Hence, it is recommended to use heuristic TSP solvers that have lower computational complexity such as nearest neighbor (Johnson & Mc-Geoch, 1997), greedy (Johnson & McGeoch, 1997), Christofides (Christofides, 1976) and etc.

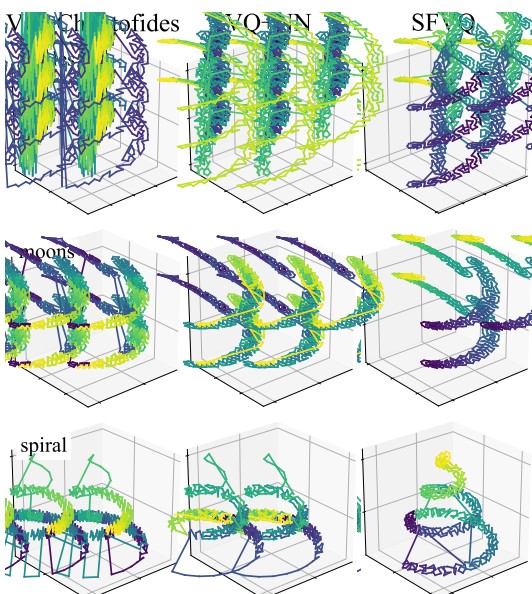

Figure 9: Comparison of codebook arrangement property of the SFVQ with ordinary VQ post-processed by *traveling salesman* solvers over three sparse distributions.

To compare the performance of TSP heuristic solutions with the SFVQ, we examine their ability to model three sparse toy distributions of *circles*, *moons*, and *spiral* in 3D space. We chose the distributions to be sparse because it makes the task more challenging. We applied ordinary VQ and SFVQ with different bitrates ranging from 7 to 10 bit (with identical initialization and hyper-parameter settings), and then for VQ we rearranged its codebook vectors by the nearest neighbor (NN) and Christofides TSP heuristic solvers. Figure 9 demonstrates the results for the 9 bit case such that the order in the space-filling line is shown with color coding (light to dark color = first to last codebook vector) for both methods of VQ+TSP and SFVQ. Since the training objective of SFVQ is different from VQ, SFVQ locates the codebook vectors such that the line connecting the subsequent codebook vectors mainly desires to fill up the distribution space and as a result, the line ends up landing inside the distribution space. To affirm this fact, look at the upper and lower parts of *spiral* dataset arranged by VQ+Christofides and VQ+NN methods. VQ locates fewer codebook vectors for these two parts of the *spiral* data, and thus we observe a narrow line that does not fill the distribution's space properly. Furthermore, we notice more unfavorable jumps (outsider lines or lines breaking the arrangement) for VQ+TSP methods than the SFVQ, due to their improper codebook arrangement. Therefore, we generally observe that the SFVQ achieves a much better codebook arrangement than VQ+TSP for all three distributions. The property of having lines mostly located inside the distribution's space is desirable for controllable data augmentation which we discuss in section A.4 in the Appendix.

## 6 CONCLUSION

Generative adversarial networks (GANs) are well-known image synthesis models that are widely used to generate high-quality images. However, still there is not sufficient control over generations in GANs, because their latent distribution acts as a black-box and is thus hard to interpret. In this paper, we used the novel unsupervised space-filling vector quantizer (SFVQ) technique to get a universal interpretation of the latent distribution of GANs as well as to find their interpretable directions of changes. Our experiments showed that the SFVQ is capable of capturing the underlying morphological structure in the latent space and discovering better and more consistent interpretable directions compared to the state-of-the-art GANSpace method. SFVQ gives the user a proper control for generating images and manipulating them, and reduces the search space for finding the desired direction of a change. SFVQ is a generic tool for modeling distributions that is neither restricted to any specific neural network architecture nor any data type (e.g. image, video, speech, etc.).

## 7 REPRODUCIBILITY STATEMENT

In our GitHub repository, we uploaded the PyTorch code of the space-filling vector quantizer (SFVQ) along with an example of how to train it on a sample Gaussian distribution. In addition, we put some other Python codes which give the instructions on how to track the training logs manually and make sure that SFVQ is getting trained in the correct way, how to initialize the codebook vectors for SFVQ, and how to expand the SFVQ's bitrate (number of codebook vectors) at each recursion step. We also uploaded the discovered directions of Figure 2 and Figure 6 along with a code that makes the user capable of applying and experimenting with these directions. Furthermore, we uploaded the learned SFVQ's codebook vectors for different datasets over different bitrates for StyleGAN2 and BigGAN models along with a code to plot the corresponding images from these codebooks. We also put the code for controllable data augmentation (discussed in section A.4 in the Appendix). At the end, in the `README.md` file we mentioned all other relevant details such as required Python libraries and how to train the SFVQ together with other modules that require gradient.

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

## A  APPENDIX

### A.1  STYLEGAN2: UNIVERSAL INTERPRETATION (CONTINUATION)

Similar to what discussed in section 4.1, we apply the SFVQ to capture a universal morphology of the latent distribution and we expect that consecutive codebook vectors in SFVQ refer to similar images. Hence, we applied a 6 bit SFVQ on the $\mathcal{W}$ space of the StyleGAN2 model pretrained on AFHQ dataset. Images corresponding to the SFVQ's codebook vectors are represented in Figure 10. We can see that similar animal species are generally located next to each other. In addition, there are some other similarities among neighboring codebook vectors such as change in rotation (from right to left) when moving from index 0 to index 10, change in rotation (from left to right) when moving from index 26 to index 34, light-colored animals for indices [22-25], bi-colored animals for indices [26-29], and baby-aged cats for indices [61-62].

### A.2  CLASS-AGNOSTIC DIRECTIONS FOR STYLEGAN2 PRETRAINED ON AFHQ

According to what discussed in section 4.2, in this section we aim to test whether and how the discovered direction of *Bicolor Face* (in Figure 2) is class-agnostic across different AFHQ animal classes. To this end, we applied this direction to all existing animal species in AFHQ dataset and represent the results in Figure 11. We observe that this direction works well for *Cat* and *Dog* classes, because there exist enough data (i.e. cats and dogs with bicolored face) within AFHQ dataset. Therefore, the learned latent space supports this transformation. In addition, this transformation

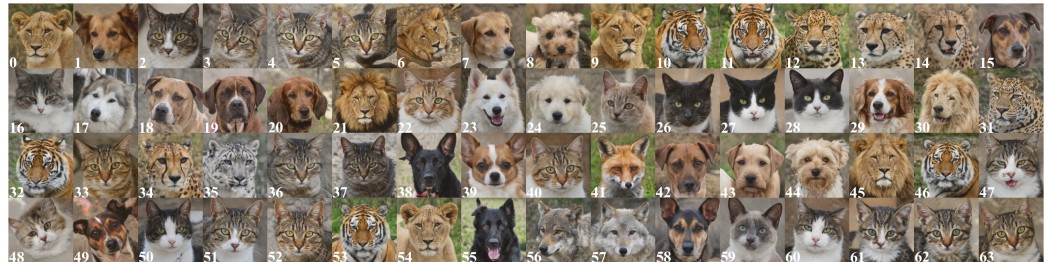

Figure 10: Codebook vectors of a 6 bit SFVQ in $\mathcal{W}$ space of StyleGAN2 pretrained on AFHQ.

more or less works for *Wolf* class, since *Wolf* looks like Siberian husky (which exists in AFHQ dataset) and this transformation leads the *Wolf* class to become similar to a Siberian husky. However, the *Bicolor Face* direction does not work for other animal classes of *Fox*, *Leopard*, *Cheetah*, *Tiger*, and *Lion*. The reason comes from the fact that the learned latent space is constrained by dataset bias of individual classes (Jahanian et al., 2019). In other words, since there is not any image with bicolored face from these animal classes within AFHQ dataset, the learned latent space does not support this transformation for them. The $\sigma$ value determines the magnitude of step which we take toward *Bicolor Face* direction of change. To make sure whether this direction works for these five animal classes, we used larger $\sigma$ value (bigger steps) for these classes. We observe that even with larger steps, not only there is no meaningful transformation effect of the desired direction, but also in the very last step ($3\sigma$) the images turn to become unrealistic by having some artifacts.

### A.3 CLASS-AGNOSTIC DIRECTIONS FOR BIGGAN PRETRAINED ON IMAGENET

As discussed in section 4.3, we found that the discovered directions (in $p(\mathbf{z})$ space of BigGAN) by SFVQ for the *golden retriever* class are class-agnostic. It means the detected directions also work when applied on other data classes within the ImageNet dataset. To confirm this fact, we applied all five directions found for the *golden retriever* (in Figure 6) on the *husky* class, and we illustrate the results in Figure 12. The image in the middle column (in red square) is the initial test image to which we apply the directions, such that we step along both sides of a direction. According to the figure, we observe that all five directions are valid for *husky* class resulting in meaningful and expected transformations.

### A.4 CONTROLLABLE DATA AUGMENTATION

According to what discussed in section 5, having lines mostly located inside the distribution's space and rare unfavorable jumps makes the SFVQ desirable for data augmentation task, such that we have a large number of meaningful latent points (located on the SFVQ's curve) to generate valid images by GANs. By plotting SFVQ's corner points as in Figure 10, we have an idea of potential

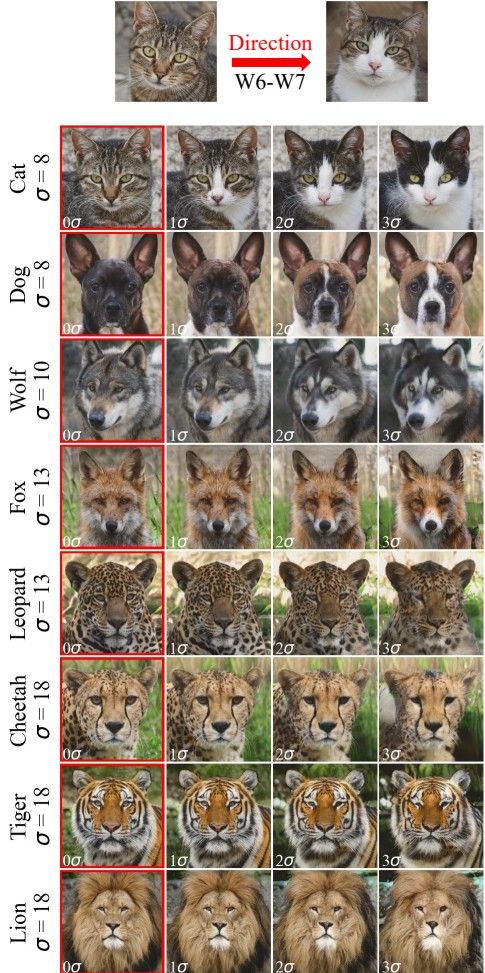

Figure 11: Applying *Bicolor Face* direction to different animal species of AFHQ dataset.

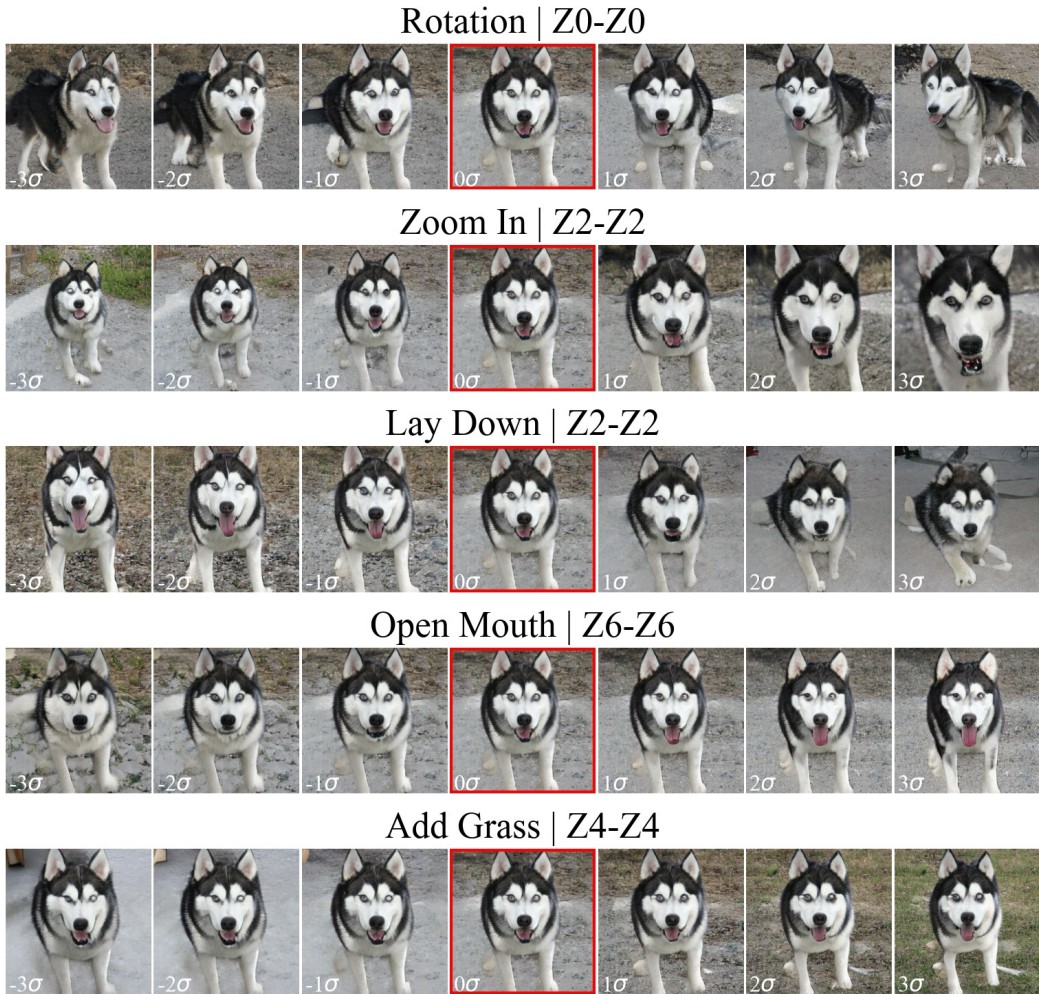

Figure 12: Class-agnostic directions; applying five discovered directions by SFVQ for the *golden retriever* class on the *husky* class using BigGAN pretrained on ImageNet.

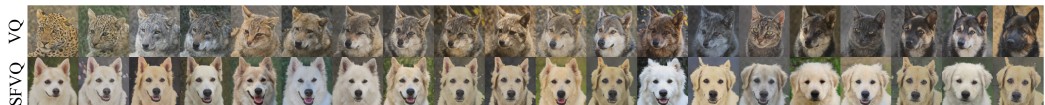

Figure 13: Generated images from 20 equally-spaced points on the line connecting two neighboring codebook vectors of VQ and SFVQ for AFHQ dataset.

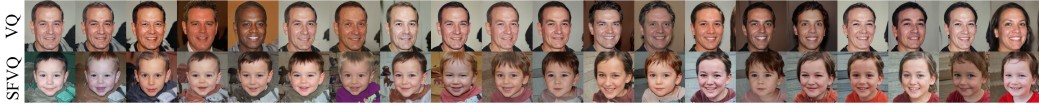

Figure 14: Generated images from 20 equally-spaced points on the line connecting two neighboring codebook vectors of VQ and SFVQ for FFHQ dataset.

generations from each SFVQ's line in advance. Hence, we can have a controllable data augmentation by generating $N$ new images while taking $N$ points on the line connecting two subsequent codebook vectors. By *controllable* we mean, the user has control over what type of images with specific characteristics he/she intends to augment by selecting the corner points from the visualized SFVQ curve. Accordingly, we chose the subsequent codebook vectors of index 23 and 24 from

Figure 10, and then we took $N = 20$ equally spaced points on the line connecting them. To have more diverse generations, we added noise to the selected points in random orthogonal directions to the line. Besides, we did the same generations with the same setting for a 6 bit VQ as well. The generations for these $N = 20$ points are demonstrated in Figure 13. As expected, all generations of SFVQ follow properties of their corner points consistently such that they are all from the same breed with light color and change in age and rotation. However, the generations for two neighboring codebook vectors (under Euclidean distance) of VQ do not follow any specific rule as their rotation, color, and breed change several times.

There are some cases in which two neighboring codebook vectors of VQ can give more consistent generations, but such cases are not mainly expected due to the different properties of VQ compared to SFVQ. However, we mainly have a consistency in generations when sampling between two neighboring codebook vectors of SFVQ. As another experiment, we also sampled $N = 20$ equally spaced points from the line connecting two baby-aged corner points (codebook vectors) of a SFVQ trained $\mathcal{W}$ space of StyleGAN2 for FFHQ dataset. The generated images are shown in the bottom row of Figure 14. We also take two neighboring codebook vectors (under Euclidean distance) of a 6 bit VQ trained on the same $\mathcal{W}$ space as SFVQ, and plot the generations in the top row of Figure 14. We observe that for SFVQ we get a good consistency in generations as all 20 generated images are from baby-aged faces. However, the generated images for VQ does not follow a specific rule as we observe changes in gender, age, face color among them.

