# OpenReview forum: "Interpretable Latent Distributions Using Space-Filling Curves"
_ICLR.cc/2024/Conference — Submitted to ICLR 2024_

### Official Review · Reviewer_Bd8w · 2023-10-24

**Soundness:** 2 fair
**Presentation:** 2 fair
**Contribution:** 1 poor
**Rating:** 3
**Confidence:** 4

**Summary:**

The authors have applied the method introduced in (Vali and Backström, 2023) to the intermediate latent space of Generative Adversarial Networks (GANs) used for generating image data. This method aims to disentangle the latent space of generative models by learning a set of vectors (codebook) that are designed to quantize a space-filling curve. The experiments mainly involve visualizing the images generated by the vectors from the learned codebook.

**Strengths:**

* The visualizations show interesting properties of the latent-space filling curve, with images from same classes being clustered at nearby indexes in an unsupervised manner.

**Weaknesses:**

* Lack of Novelty and Originality: The paper primarily focuses on the application of an existing method to new models and datasets, without introducing any novel or original contributions. There is no apparent effort to adapt or modify the method to suit the characteristics of the new models and dataset. In my view, this lack of novelty and originality raises questions about whether this paper is suitable for submission to ICLR, as it could be perceived as a dual submission.

* Insufficient Experimental Rigor: The experimental setup in the paper is notably lacking in terms of rigor. The absence of quantitative metrics and reliance solely on visualizations is a significant shortcoming. Without comprehensive metrics, the results can be susceptible to cherry-picking and fail to provide a strong empirical validation of the method.

**Questions:**

-

**Details Of Ethics Concerns:**

The paper reuses the method from the following paper: https://www.isca-speech.org/archive/pdfs/interspeech_2023/vali23_interspeech.pdf

There is no change in the method, but they only apply it on another dataset and models.
I have concerns that it might be considered as a dual submission.

Note that the authors cite the given paper and do not try to "hide" it.

---

> ### Author Response · Authors · 2023-11-14
> **Answers to comments in Weaknesses section**
>
> 1. Thanks for your invaluable comments! The SFVQ presented in  (Vali and Bäckström, 2023) was originally used for universal interpretation of the underlying morphological structure of the latent space (or any distribution). However, in this paper apart from those universal interpretations, we discovered that in a SFVQ's curve the line connecting two corner points can work as an interpretable direction for changing attributes. Also we noticed that SFVQ works similar to PCA as its lines lies along the directions in which the data has the most variance (Fig. 5), but its main difference with PCA is that SFVQ's lines are not orthogonal to each other. According to this property, we can state why  in Figure 7 our "Gender" direction remains more in the valid range than PCA-based GANSpace, in which "Gender" and "Rotation" are orthogonal directions to each other.
> In addition, we showed that SFVQ can be considered as a sorted VQ method which can perform better than traveling salesman problem in sorting especially for sparse data (Section 5 and Figure 9). According to section 5, we also showed that we can use the huge number of data points lied on the SFVQ’s curve as meaningful points for controllable data augmentation (Fig 13 and Fig 14 in Appendix). These are all new things we found on top of the original SFVQ paper. Furthermore, according to Figure 2, we discovered 13 different directions for FFHQ dataset, which is more than the number of directions found in most of the state-of-the-art papers. Also, some directions are really specific and unique to our proposed method such as “Asian Face”, “Hat”, “Graduation Hat”, “Beard” for human faces (FFHQ), “Age” for animals (AFHQ).
>
> 2. About the quantitative metric, thanks to the recommendation from one of the reviewers, I am going to apply the quantitative measures used in [1] (commutativity error, side effect error, and identity error) and compare our discovered directions with some existing state-of-the-art methods. Unfortunately, I think I cannot complete the experimentation before the rebuttal period ends as it takes some time.
>
> [1] Aoshima, Takehiro, and Takashi Matsubara. "Deep Curvilinear Editing: Commutative and Nonlinear Image Manipulation for Pretrained Deep Generative Model." Proceedings of the IEEE/CVF Conference on Computer Vision and Pattern Recognition. 2023.

---

> > ### Comment · Reviewer_Bd8w · 2023-11-20
> >
> > I appreciate the authors' response. However, I maintain my initial rating (3: Reject) due to persisting concerns.
> >
> > **Lack of novelty**: I respectfully disagree with the authors. My contention remains that utilizing an existing method on a different model and dataset without substantial adaptations lacks the necessary depth for a strong contribution. The resemblance to the analysis in (Vali and Bäckström, 2023), demonstrating the grouping of similar data by nearby codebook vectors, undercuts the novelty. Additionally, the introduced contributions, such as the link with PCA and attribute modification via directions, seem rather direct. Furthermore, as previously highlighted, these contributions heavily rely on experimental observations validated solely through visualizations, which I consider of limited scientific value.
> >
> > **Experimental rigor**: My concerns persist as the paper hasn't addressed this aspect.

---

### Official Review · Reviewer_XT3P · 2023-10-30

**Soundness:** 2 fair
**Presentation:** 1 poor
**Contribution:** 1 poor
**Rating:** 3
**Confidence:** 5

**Summary:**

The paper presents a method for discovering interpretable distributions in the latent space of pre-trained GANs (StyleGAN2 and BigGAN) using the previously proposed Space-filling vector quantizer (SFVQ). The main intuition of the paper is that the corner points of the space-filling curves (i.e., VQ codebook elements), which is expected to capture the structure of the latent space, will refer to similar content, and thus will provide a path to control/interpret the generation process. The proposed method is compared qualitatively with GANSpace, an unsupervised method for discovering interpretable direction using PCA, on two GAN architectures (StyleGAN2 and BigGAN).

**Strengths:**

The idea of using space-filling curves in the context of pre-trained GANs, in order to model interpretable/controlable generative paths is interesting.

**Weaknesses:**

Whilst I find the idea of using the Space-filling vector quantizer (SFVQ) for discovering interpretable/controlable latent paths in pre-trained GANs interesting, the paper fails to convince why this is an effective approach towards that goal.

First and most importantly, the paper does not discuss or compare with two very relevant recent works [1, 2]. These works are both unsupervised and model-agnostic, and, similarly to GANSpace, should necessarily be included in the experimental evaluation of the proposed method.

Second, the presented empirical evaluation is only qualitative, rendering impossible to compare its advantages against GANSpace (and other relevant works that are missing [1, 2]). Even in that case, in the only figure that provides comparison with an existing work (i.e., Fig. 7, comparison with GANSpace), it is hardly visible why the proposed method is better.

In absence of any quantitative comparison, it is impossible to assess the merits of the proposed method. Appropriate quantitative metrics can be found in [1, 2]. In my point of view, the paper should compare with those methods using the suggested metrics.

Finally, I found the structure of the paper hard to follow. Specifically, Sect. 4 appears to be a few unstructured blocks of text discussing limited empirical results. Also, in many occasions, the paper makes unsupported or inaccurate claims. For instance, "our unsupervised proposed method neither needs any human labeling nor puts any constraint on the learned latent distribution" is not true according to Sect. 4.2 (for instance).


[1] Aoshima, Takehiro, and Takashi Matsubara. "Deep Curvilinear Editing: Commutative and Nonlinear Image Manipulation for Pretrained Deep Generative Model." Proceedings of the IEEE/CVF Conference on Computer Vision and Pattern Recognition. 2023.
[2] Tzelepis, Christos, Georgios Tzimiropoulos, and Ioannis Patras. "Warpedganspace: Finding non-linear rbf paths in gan latent space." Proceedings of the IEEE/CVF International Conference on Computer Vision. 2021.

**Questions:**

In Sect. 4.1 you mention: "When generating W’s latent vectors during training, StyleGAN2 asks for class label of the image."
How this is the case? StyleGAN2 is not a conditional GAN, that class label is not given during generation.

Have you considered other latent space, such as W+ or the so-called style space S? I think it would be interesting to do so, since:
i) W+ can be split into 18 512-dimensional sub-spaces, where learning a SFVQ for each sub-space (layer) could lead to further controlability/interpretability, and
ii) the style space S is remarkably disentangled (changing a single element/dimension could lead to a very specific and disentangled change, such as gaze movement).

---

> ### Author Response · Authors · 2023-11-14
> **Answers to comments in Weaknesses section**
>
> 1. Thanks for your invaluable comments! We read these papers and it sounds interesting and we are excited to measure the performance of our interpretable directions regarding the quantitative measures in [1] (commutativity error, side effect error, and identity error) and compare it with other works. However, it seems that the implementation of [1] is not provided. So, we will do the comparisons with [2] and maybe other existing works. Unfortunately, I think I cannot complete the experimentation before the rebuttal period ends as it takes some time.
>
> 2. About the comparison with GANSpace (Fig. 7), those 5 directions are the only directions provided by GANSpace for StyleGAN2 on FFHQ dataset that we could use for qualitative comparisons. About the performance comparison, among the first three directions (Gender, Rotation, Smile) which are the most prevalent directions in the papers in the literature, we can easily say that regarding Gender our direction is more valid or has higher validity length than GANSpace. Also regarding Smile our direction make more sensible changes.
>
> 3. About this claim “our unsupervised proposed method neither needs any human labeling nor puts any constraint on the learned latent distribution”. As we use the pretrained networks and do not retrain them, our method does not incur any changes to the latent space of the GANs. Also, we mentioned in the introduction that our method only needs one time observation to discover directions. So, it does not need huge amount of data labeling for training. Furthermore, in references of [1] and [2] and also similarly Voynov[3], they need hyper-parameters tunings (tuning coefficients in loss function) and also architecture design for the warping (transformation) and reconstruction neural networks. All these might effect the performance of their methods, as our SFVQ training is so simple without any hyper-parameter tuning and network architectural design.
>
> [3] Voynov A, Babenko A. Unsupervised discovery of interpretable directions in the gan latent space. In International conference on machine learning 2020 Nov 21 (pp. 9786-9796). PMLR.

---

> > ### Comment · Reviewer_XT3P · 2023-11-22
> >
> > I would like to thank the authors for their responses.
> >
> > Unfortunately, they factually do not address my concerns. The lack of proper experimental evaluation, using quantitative metrics that have been used by similar existing works, and comparisons to the said works, render the evaluation of this work impossible. Whilst I understand that the discussion time might not be enough for an extensive experimental evaluation and comparisons with all the relevant works, at least some initial results would be necessary for considering changing my rating for this paper at this stage.
> >
> > Finally, important concerns raised by other reviewers, mainly regarding the novelty/originality of the proposed method, have also not been addressed.
> >
> > After reading the rest of the reviews and the authors' responses to them and to my concerns, I maintain my initial rating (3: Reject).

---

> ### Author Response · Authors · 2023-11-14
> **Answers to comments in Questions section**
>
> About your question; We downloaded the pretrained model from the official repository of Stylegan2 (on NVIDIA website under this link https://catalog.ngc.nvidia.com/orgs/nvidia/teams/research/models/stylegan2). To generate an image, it always ask for a class variable which we put “None” for datasets which do not have specific labels like FFHQ, AFHQ, and we put class label as an integer value for CIFAR10 dataset.
>
> In StyleGAN2, for an input sampled noise vector Z, we found its corresponding style vector W. And this W vector is the same for all 18 different blocks. You might not noticed in the paper, but we mentioned that we applied our SFVQ on the W (Style space) of StyleGAN2 and we find the interpretable directions in this space. We did the same experiments as GANSpace did. In our method, after finding an interpretable direction, among 18 different style vectors (Ws) we only change some W vectors (along the interpretable direction) which are the most relevant ones for changing the desired style (or attribute). That is why we claimed that for our method we can simply find the interpretable direction by simple observation (we have prior knowledge about the direction), and we only have to search for the desired style blocks to modify. However, in GANSpace there is no prior knowledge about the discovered directions and we have to search also among them to find the desired direction (apart from searching for the desired style blocks to modify). In this way our method reduced the search space (or search effort) for finding the desired direction. Do you mean this space we already experimented on?

---

### Official Review · Reviewer_SucH · 2023-11-02

**Soundness:** 3 good
**Presentation:** 3 good
**Contribution:** 3 good
**Rating:** 5
**Confidence:** 3

**Summary:**

This paper addresses the task of unsupervised learning of an interpretable latent space in GANs using space-filling vector quantization. The main idea is to model the latent space of pretrained GANs as a piece-wise continuous and linear curve anchored using a set of codebook vectors. The authors apply the proposed method to two different GAN architectures: StyleGAN and BigGAN. The proposed method is evaluated on different datasets, and a thorough analysis has been provided.

**Strengths:**

-- The paper is well-written and well-presented.

-- Although learning an interpretable latent space using space-filling vector quantization has been introduced in another previous work, its application to GANs is novel and interesting.

-- The proposed method can be applied to pre-trained GANs and does not require retraining the whole model.

-- The proposed method does not require a pre-defined number of semantic directions to be discovered.

-- The experiments are extensive, evaluating the method on two GAN models and 4 different datasets.

-- Based on the experiment, the proposed method seems effective in discovering meaningful semantic directions in the latent space

**Weaknesses:**

-- The provided results for ImageNet (BigGAN) are limited. I could only find two examples limited to the Dog category. It would be interesting to see more visualizations of how the method works on other categories, especially objects.

-- The author has not discussed and evaluated how the latent-space inversion could be performed in the proposed latent space, and whether the discovered directions are meaningful and effective for the inverted latent code.

-- I could not find an ablation on the number of the codebook vectors. I would appreciate it if the authors could further clarify this.

-- I would assume sampling only on the obtained curve could affect the image quality and diversity quite a lot. Currently, no evaluations (e.g. FID scores) are provided for the generation quality.

**Questions:**

Please see the weaknesses.

---

> ### Author Response · Authors · 2023-11-14
> **Answers to four comments in Weaknesses section**
>
> 1. Thanks for your invaluable comments! We will also find some useful directions for other BigGAN classes (like an object) and update the paper with the new directions for sure.
>
> 2. We were aware of some methods which experimented their directions in GAN inversions. However, we have not experimented the performance of our discovered directions in GAN inversions, since the performance would be dependent on the accuracy and efficiency of the inversion model as well. We will consider this case and might experiment the performance of our method in GAN inversions.
>
> 3. If we apply the SFVQ with 2 bits, we can only consider 3 different directions (as there are 3 lines connecting the 4 corner points). Hence, the more the bitrate is, the more diverse and unique directions we could find by plotting and observing generations from corner points. With lower bitrates (like 2 bits in figure 5) the learned SFVQ finds the most fundamental directions in which the data has the most diversity along these directions in the latent space like Rotation and Gender. Therefore, the directions in figure 2 are found by observing the learned SFVQ (only once) over different bitrates ranging from 2 to 12 bits.
>
> 4. As the main objective of the paper was improving the interpretablity of the latent space, we did not study the SFVQ’s generation quality. However, according to section 5, the SFVQ’s curve lies mainly inside the data distribution. Hence, if the we sample points from the SFVQ’s curve, the corresponding generated images are meaningful with good quality. Regarding the diversity, if we sample the points existed on the line between two corner points, we would not get diverse generations, as the points exhibits the transition from image A to image B, which are the corner point images. However, if we add noise to the sampled points from SFVQ’s curve (figures 13 and 14 in the Appendix) we could get more diverse and meaningful generations.

---

> > ### Comment · Reviewer_SucH · 2023-11-20
> >
> > I appreciate the author's response to my questions. However, my concerns remain unresolved:
> >
> > 1. No additional visualizations on BigGAN is provided
> > 2. How the inversion works with the proposed method is not addressed
> > 3. A more structured ablation using different bitrates is needed. I would suggest dedicating a separate section in the paper to showing the effect of choosing different bitrates
> > 4. No quantitative evaluation of the quality and diversity of the generated images using the proposed method is provided.

---

### Meta-Review · Area_Chair_c4r7 · 2023-12-14

**Metareview:**

This paper applies an existing method, [Vali and Backström, 2023], for interpreting latent representation spaces via space-filling curves, to StyleGAN2 and BigGAN.  After the author response and discussion phase, all reviewers favor reject, with concerns including lack of novelty and insufficient quantitative experimental validation.

Reviewer Bd8w raises an additional concern about the degree of overlap between the sections of text that present the method in this paper and the presentation in [Vali and Backström, "Interpretable Latent Space Using Space-Filling Curves for Phonetic Analysis in Voice Conversion", Interspeech 2023].

Even setting aside this potential issue, the paper, which cites [Vali and Backström, 2023], is an application of this existing method to other models and datasets, and as a result, Reviewers XT3P and Bd8w both believe there is not a sufficiently novel or original contribution.  All three reviewers additionally note various weaknesses in experimental evaluation.  The AC agrees with the overall reviewer consensus.

**Justification For Why Not Higher Score:**

Lack of novelty and insufficient experimental validation.

**Justification For Why Not Lower Score:**

N/A

---

### Decision · Program_Chairs · 2024-01-16

Reject